# Effect of Desert Sand on the Section Bonding Properties of Polyethylene Fiber−Engineered Cementitious Composites

Yanfeng Niu [1], Fengxia Han [1,2,*], Qing Liu [1,2] and Xu Yang [1]

1  College of Architectural and Civil Engineering, Xinjiang University, Urumqi 830047, China; yanfeng@xju.edu.cn (Y.N.)
2  Key Laboratory of Building Structure and Seismic Resistance of Xinjiang, Urumqi 830017, China
*  Correspondence: fxhan@xju.edu.cn

**Abstract:** Xinjiang is in northwest China and has abundant desert sand. Replacing natural sand with sand from deserts is an urgent need and could be used in making polyethylene fiber−engineered cementitious composite (PE−ECC). The interfacial bonding properties of desert sand PE−ECC (DSPE−ECC) were made using the optimal mix proportion (30% desert sand content, 2% fiber volume) and the laboratory's previous research results. Normal sand PE−ECC (NSPE−ECC) and DSPE−ECC at different test ages (3, 7, 14, and 28 days) were subjected to uniaxial tensile tests, and a method for determining bonding properties is proposed. Scanning electron microscopy and X-ray diffraction were used to analyze the development of PE−ECC fiber and matrix and the formation of hydration products. The results indicated that the cracking loads of the DSPE−ECC at 3, 7, 14, and 28 days increased by 16.72%, 28%, 23.23%, and 10.05%, respectively. Desert sand had low water content and high water absorption, which slowed down the rate of $C_2S$, $C_3S$ combining with water molecules to form C−S−H, and had a great influence on the bonding properties of ECC at 3 days. However, the bonding properties of DSPE−ECC were only slightly less than those of NSPE−ECC at 28 days, and the bonding properties had gradually stabilized. Therefore, the addition of desert sand enhanced the fiber/matrix's bonding properties, and the bonding properties stablized with the increase in curing ages.

**Keywords:** desert sand; interfacial bonding properties; C−S−H gel; polyethylene fiber; engineered cementitious composite

## 1. Introduction

With the fast growth of civil engineering infrastructure, concrete has become a popular building material for foundations. Many researchers have studied concrete's mechanical properties and durability [1–4]. Concrete consists of fine aggregate, coarse aggregate, cement, water, mineral admixtures, and chemical additives. Fine and coarse aggregates are the main parts of concrete, making up 70%–80% of the volume of concrete [5]. Most sand comes from natural rivers. The use of river sand has caused severe environmental problems. Moreover, improving the performance of concrete is essential [6–8]; therefore, looking for alternatives for fine aggregate is an urgent need. Northwest China has 80% of China's desert resources [9–11], with Xinjiang, Inner Mongolia, Gansu, and Ningxia being the most important regions. Using desert sand instead of natural river sand can help the environment in some ways and solve the lack of building materials. Researchers have made much progress in their study of how desert sand can be used [12–17].

In engineering, the ability of concrete to bond between surfaces is essential [18,19]. Concrete has been used in many practical projects because it is cheap and easy to make [20]. However, concrete has low tensile strength, is heavy, cracks easily, and has poor durability properties [21,22]. Hence, these characteristics make some engineered concrete structures vulnerable to damage [23,24]. In the past 20 years, researchers have studied engineered

cementitious composite (ECC) with ultra−high toughness [25–31] to find ways to make concrete stronger, more flexible, and less likely to crack. Steel fiber, polybutylene alcohol (PVA) fiber, polybutylene (PE) fiber, and hybrid fibers are standard ECC fibers. PE−ECC and PVA−ECC can withstand more than 3% tensile strain without breaking. PVA fiber is inexpensive; therefore, more research has been done on PVA−ECC [32,33]. However, at the same volume dosage, the strain capacity of PE−ECC is slightly larger than that of PVA−ECC, and the friction coefficient of PE fiber is higher than that of PVA fiber. Additionally, PE fiber repels water; thus, it will not change how ECC cement hydrates and will strengthen the bond between cement and fiber. When fibers are added to cement−based materials, they can make up for the fact that cement−based fabrics are fragile and do not have much tensile strength. The main reason to improve the bond between the cement matrix and the fiber is to make cement−based materials stronger [34]. Researchers have learned much about how ECC bonds at the interface [35–38]. Previous results show that ECC can improve how well concrete bonds at the joints and can be used as a long−lasting repair material for structures. However, how ECC fiber and matrix bond at the interface when desert sand is used instead of natural sand needs more research. If desert sand is used to replace part of natural sand to prepare ECC and can meet the engineering requirements, the use of natural resources can be reduced to a great extent, and the application of ECC can be more extensive.

Experiments were conducted by researchers to assess the bridging effect and toughness of ECC fiber. For example, Xuan et al. [39] used scanning electron microscopy (SEM) to examine the interface bonding mechanism of fiber−reinforced cementitious composite on a small scale and showed that the composite's bonding strength was higher than the tensile strength of the cement matrix. Abousnina et al. [40] found that when the volume fraction of polypropylene fiber is 2%, the fiber plays the most obvious bridging role, and the toughness index of the concrete is the best, which leads to progressive ductile failure. Many researchers also used the combination of numerical simulation and experiments to evaluate the interface bonding properties of ECC. Cui et al. [41] used molecular dynamics simulations and atomic force microscopy to determine how well asphalt and aggregate stick together at the interface. Jin et al. [42] used digital image correlation and molecular dynamics simulation to test how well fibers and epoxy bond at the interface. Currently, there are few theoretical ways to measure the interfacial bonding properties and toughness of ECC and no standard procedures have been established. The interaction between the fibers and cement matrix in ECC can make materials much harder to break because fiber bridging stops microcracks from forming in the cement matrix and takes up the tension across cracks [43,44]. The role of bridging is determined by how much energy is absorbed when fibers are pulled out of the cement matrix [45]. Therefore, in this paper, we suggest a way to evaluate the section bonding performance of ECC from a theoretical point of view based on the stress formula and the fracture mechanics energy criterion.

The bonding properties between fiber and matrix have a great influence on the tensile properties of ECC. This paper reports the effect of desert sand replacing 30% natural sand on the bonding properties between fiber and matrix. The uniaxial tensile properties of desert sand PE−ECC (DSPE−ECC) were compared with natural sand PE−ECC(NSPE−ECC) as the control group using uniaxial tensile tests on specimens at different curing ages. We propose a theoretical method for evaluating the bonding properties of the DSPE−ECC fiber and matrix at the interface. Moreover, SEM and XRD were used to study the bonding properties of NSPE−ECC and DSPE−ECC.

## 2. Materials and Methods

### 2.1. Material Properties and Mix Proportion Design

The specimens were cast with P·O 42.5 cement, grade II fly ash natural sand with a particle size of less than 1.18 mm, Kumtag desert sand, and PE fiber. The chemical composition of the desert sand used in the study is listed in Table 1. Sand water absorption tests were carried out according to the operational specifications [46]. The average water

absorption of desert sand was 1.1% and that of natural sand was 0.75%. The main water absorption test results of sand are shown in Table 2.

**Table 1.** Chemical composition of desert sand.

| Composition | $SiO_2$ | FeO | $Fe_2O_3$ | $AL_2O_3$ | MgO | $K_2O$ | CaO | $Na_2O$ |
|---|---|---|---|---|---|---|---|---|
| Content/wt% | 82.66 | 0.80 | 1.05 | 8.72 | 1.51 | 0.12 | 2.00 | 0.07 |

**Table 2.** Water absorption of sand.

| Type of Sand | Test 1/% | Test 2/% | Average Value/% |
|---|---|---|---|
| Desert Sand | 1.0 | 1.2 | 1.1 |
| Natural sand | 0.8 | 0.7 | 0.75 |

Particle grading tests were carried out on the sand according to the specifications, and the results showed that the natural sand particles belonged to grading zone III, with fineness modulus Mx = 1.87 and particle grading between 2.3 mm~1.5 mm, which indicates fine sand. The desert sand grain gradation zone belonged to the overfine sand zone, with fineness modulus Mx = 0.60, which indicates powder sand. The main sieving results of sand are shown in Table 3. The PE fiber was acquired from Shandong Laiwu Telif Company in China. The PE fiber composite material had a surface density of 0.97 g/cm$^3$ and a single fiber diameter of 24 μm. The tensile strength, modulus of elasticity, and elongation of the PE fiber were 3000 MPa, 120 GPa, and 5%, respectively. The main mechanical property parameters of PE fiber are shown in Table 4.

**Table 3.** Sieving results of sand.

| | Particle Size | 4.75 mm | 2.36 mm | 1.18 mm | 600 μm | 300 μm | 150 μm | 75 μm |
|---|---|---|---|---|---|---|---|---|
| | Sieve margin/g | 0 | 0 | 0 | 140.45 | 209.45 | 95.75 | 30.74 |
| Natural Sand | Subtotal sieve residual/% | 0 | 0 | 0 | 28.09 | 41.89 | 19.15 | 6.15 |
| | Cumulative sieve balance/% | 0 | 0 | 0 | 28.09 | 69.98 | 89.13 | 100 |
| | Sieve margin/g | 0 | 0 | 0 | 17.3 | 45.11 | 156.55 | 272.13 |
| Desert Sand | Subtotal sieve residual/% | 0 | 0 | 0 | 3.46 | 9.02 | 31.31 | 54.43 |
| | Cumulative sieve balance/% | 0 | 0 | 0 | 3.46 | 12.66 | 43.97 | 100 |

**Table 4.** PE fiber performance parameters.

| Density /(g/cm$^3$) | Tensile Strength /MPa | Elastic Modulus /GPa | Ultimate Elongation /% | Length /mm | Diameter /μm |
|---|---|---|---|---|---|
| 0.97 | 3000 | 120 | 5 | 12 | 24 |

In order to study the section bonding properties of desert sand, the mix was divided into 2 groups. The best PE−ECC ratio was chosen based on the results of the authors' previous research [47,48]. The first group was a natural sand group, which was used as a control group of desert sand. The second group was the desert sand experiment group, in which 30% natural sand was replaced and the volume content of PE fiber was 2%. The specific mix proportions are shown in Table 5.

**Table 5.** Mix proportion of PE−ECC.

| Specimen Number | Cement /% | Fly Ash /% | Natural Sand /% | Desert Sand /% | Water /% | Fiber /% | Water Reducer /% | Thickening Agent /% |
|---|---|---|---|---|---|---|---|---|
| NSPE−ECC | 43.95 | 18.84 | 18.84 | 0 | 16.96 | 0.90 | 0.49 | 0.02 |
| DSPE−ECC | 43.95 | 18.84 | 13.19 | 5.65 | 16.96 | 0.90 | 0.49 | 0.02 |

### 2.2. Specimen Design and Test Setup

PE−ECC test specimens were made according to the specifications [49]. The size of the specimen's outside edge was $230 \times 60 \times 15$ mm$^3$, and the middle part of the specimen was $80 \times 30 \times 15$ mm$^3$ (Figure 1). The specimens were cured for 3, 7, 14, and 28 days in the same environment (temperature: $20 \pm 2$ °C, relative humidity: $\geq 95\%$).

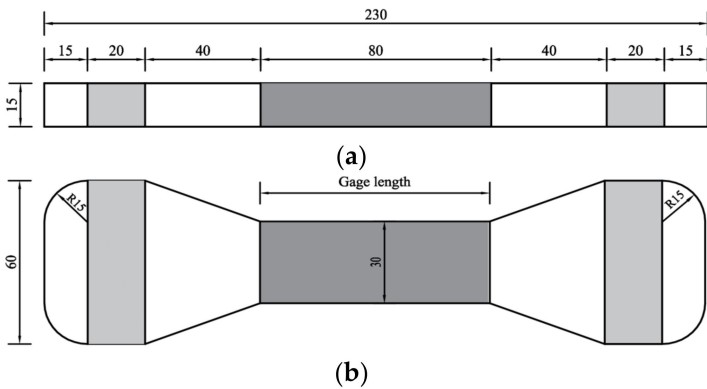

**Figure 1.** Size of specimen (unit: mm): (**a**) Side view, (**b**) Plan view.

All specimens were tested by a displacement−controlled uniaxial tensile test machine (5 KN liquid crystal electronic tension machine; model LDS−5; Jinan Chuanbai Instrument Equipment Co., Ltd. Jinan, China). The running speed of the testing machine was controlled at 0.5 mm/min for the entire test, and the data acquisition frequency was three times/s. The middle 80 mm of each specimen was selected for measurement. Two linear variable differential transformers were placed at the left and right sides of each specimen (Figure 2). Each group of specimens had the same ratio, maintenance conditions, tensile mode, and other conditions to ensure the experimental results and data were accurate and comparable.

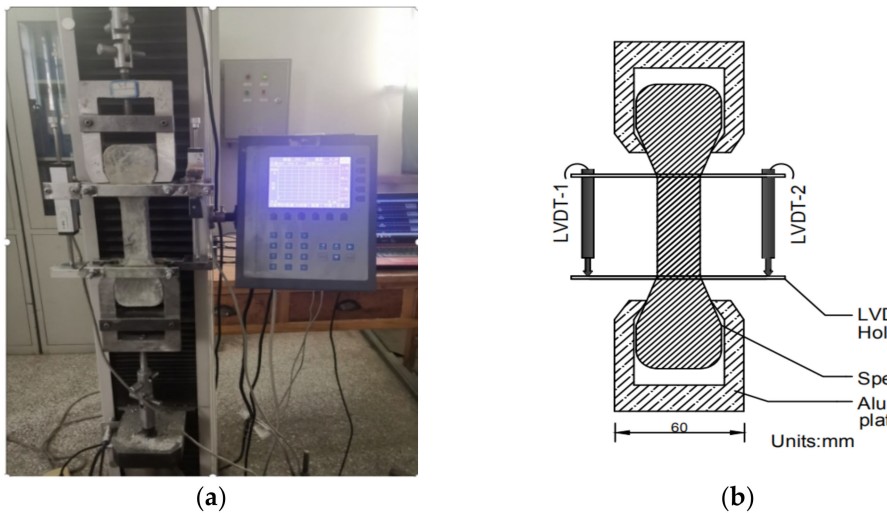

**Figure 2.** Tensile testing equipment: (**a**) loading instruments, (**b**) loading sketch.

### 2.3. Bonding Properties Evaluation Index

The bonding properties of the PE−ECC specimen's fiber and matrix were evaluated using the deformation energy set uniaxial tensile test proposed in the literature [50]. The elastic stage of PE−ECC was the first stage, the transition from the elastic stage to the strain−hardening stage was the second stage, and the strain−hardening stage was the final stage. The bonding strength and the average number of cracks were used to evaluate the bonding properties of PE−ECC. The formulas for calculating the PE−ECC's bonding properties index are shown below.

(1) Formula for bond strength calculation

The deformation energy was substituted into the stress formula, and the difference between the ultimate load deformation energy and the crack load deformation energy was used to determine the bonding effect as shown in Equation (1):

$$\tau = \frac{U_s - U_c}{\pi d_f L_f (u_s - u_c)} \tag{1}$$

where $d_f$ is the fiber diameter, $L_f$ is the fiber length, $U_s$ is the maximum deformation energy in the softening phase and the deformation energy in the third stage of the corresponding load−displacement curve, $U_c$ is the matrix cracking deformation energy, $u_s$ is the maximum displacement in the softening phase, and $u_c$ is the matrix cracking displacement.

(2) Formula for average crack number

Under the energy criterion of fracture mechanics, the average number of cracks was determined by dividing the total energy number by the energy of a crack, and the cracking effect caused by the bond failure between the fiber and matrix of the ECC tensile test piece was evaluated as shown in Equation (2):

$$n = \frac{U_s - U_{fc}}{U_{fc} + \dfrac{k u_{fc}^{2}}{2}} \tag{2}$$

where $U_{fc}$ is the total deformation energy of the first and second stages, $U_s$ is the maximum deformation energy in the softening phase, $k u_{fc}^{2}/2$ is the modification of the strain−hardening stage to the previous stage, $u_{fc}$ is the displacement of the second stage of the curve, and $k$ is the slope of curve fitting in the third stage.

## 3. Results and Discussion

### 3.1. Test Failure Phenomenon

The uniaxial tensile test was conducted to achieve the objective of the section bonding properties of the PE−ECC. Figure 3 shows the tensile failure cracks of the NSPE−ECC and DSPE−ECC specimens. Multiple cracks appeared in all specimens at certain lengths. Before PE−ECC failed in tension, no cracks formed on the surface of the test piece. When the load reached the cracking load of the test piece, several small cracks that went in different directions appeared on the surface of the test piece. Many cracks came together, and a primary crack formed quickly when the load reached the breaking point. Cracks formed rapidly and broke the specimen in half with a sound when the load reached the failure value.

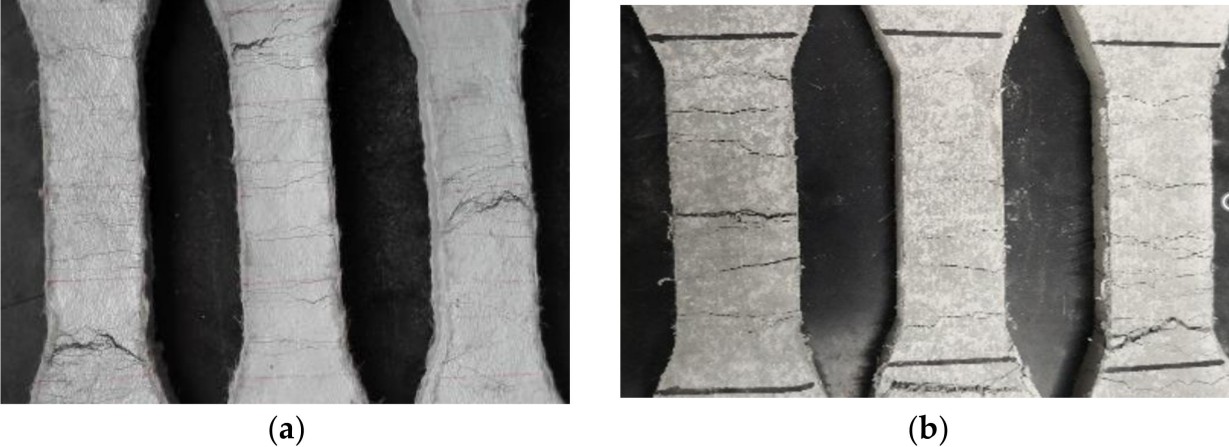

**(a)**

**(b)**

**Figure 3.** Damage to specimens: (**a**) damage to NSPE−ECC; (**b**) damage to DSPE−ECC.

Figure 4a,b shows the presence of many fibers in the failure section. This occurred because most of the bonding force of the PE−ECC comes from the connection between fibers and the matrix. The specimen failed when the fibers were pulled out of the matrix of the test piece or broken.

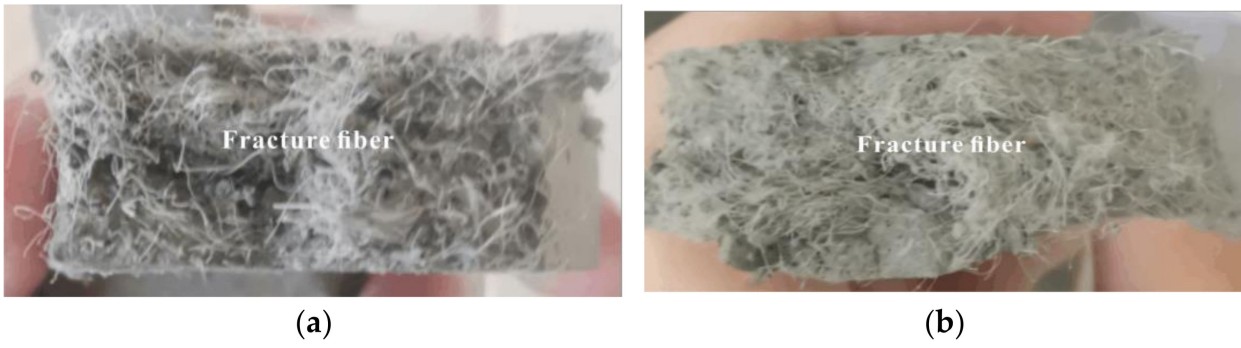

**(a)**

**(b)**

**Figure 4.** Damage section of specimens: (**a**) damage cross−section of NSPE−ECC; (**b**) damage−cross section of DSPE−ECC.

### 3.2. Load−Displacement Curve of Different−Aged Specimens

Figure 5 depicts the load–displacement curves of NSPE−ECC and DSPE−ECC specimens in the uniaxial tensile test. The fiber and matrix were subjected to the same stress during the specimen's elastic stage. When the load did not reach the matrix's cracking load, the surface remained unchanged, and the displacement was minimally altered. When the load reached the matrix's cracking load, the matrix developed a thin crack at its weakest point. The friction bond force of the fiber matrix at the section was substantially greater than the matrix's cracking load; hence, the fiber supported the tension necessary to separate the surrounding matrix or the weak area of the matrix at other locations. The formation and contraction of the matrix crack constituted a fluctuating strain−hardening stage on the load–displacement curve. Figure 5a,e shows that the cracks of DSPE−ECC at an age of 3 days were considerably less than those of NSPE−ECC and that the failure mode had poor ductility. Although the displacement of DSPE−ECC at 28 days was less than that of NSPE−ECC, it exhibited excellent multi−crack steady−state cracking and ductility, as shown in Figure 5d,h.

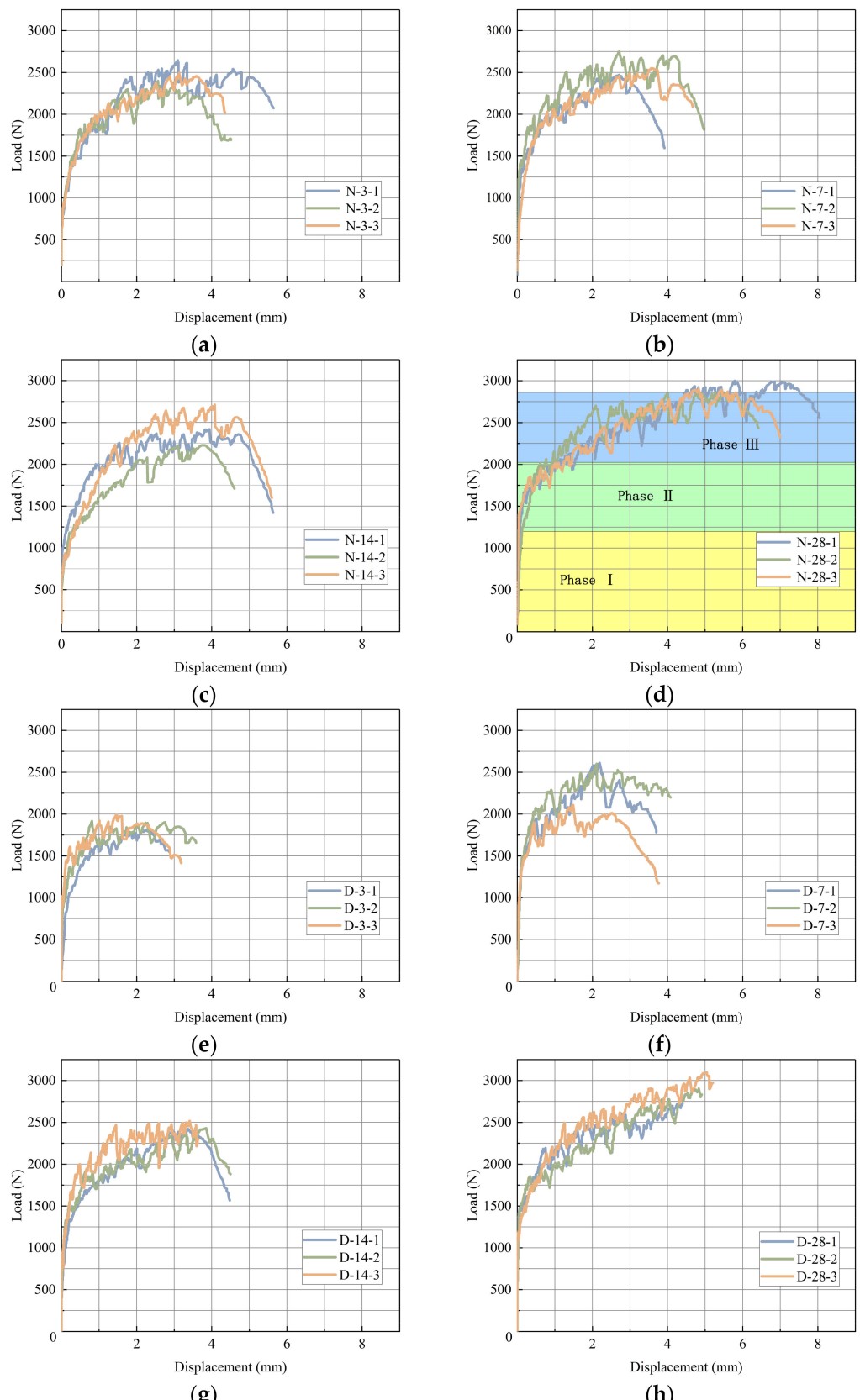

**Figure 5.** Load−displacement curves: (**a**) 3 d, (**b**) 7 d, (**c**) 14 d, (**d**) 28 d of NSPE−ECC, (**e**) 3 d, (**f**) 7 d, (**g**) 14 d, (**h**) 28 d of DSPE−ECC.

Table 6 shows that the cracking and ultimate loads increased with age. NSPE−ECC had a higher ultimate load and better tensile properties than DSPE−ECC. The ultimate load of DSPE−ECC at an age of 3 days was considerably lower than that of NSPE−ECC. However, the ultimate load of DSPE−ECC at 28 days was slightly lower than that of NSPE−ECC with increasing age. This result showed that DSPE−ECC hydrates more slowly than NSPE−ECC, which affects the tensile load of DSPE−ECC. Moreover, the cracking loads of DSPE−ECC at 3, 7, 14, and 28 days were 16.72%, 28%, 23.23%, and 10.05% higher than those of NSPE−ECC at the same age, respectively.

**Table 6.** NSPE−ECC and DSPE−ECC uniaxial tensile test results.

| Specimen Number | Cracking Load /N | Average Cracking Strength/N | Ultimate Load /N | Average Ultimate Load/N |
|---|---|---|---|---|
| N-3-1 | 849 | | 2645 | |
| N-3-2 | 899 | 899.3 | 2398 | 2512.3 |
| N-3-3 | 950 | | 2494 | |
| N-7-1 | 961 | | 2470 | |
| N-7-2 | 1227 | 1156.0 | 2749 | 2589.7 |
| N-7-3 | 1280 | | 2550 | |
| N-14-1 | 1237 | | 2418 | |
| N-14-2 | 887 | 961.0 | 2226 | 2451.7 |
| N-14-3 | 759 | | 2711 | |
| N-28-1 | 1207 | | 2999 | |
| N-28-2 | 1075 | 1227.7 | 2859 | 2920.3 |
| N-28-3 | 1401 | | 2903 | |
| D-3-1 | 1149 | | 1813 | |
| D-3-2 | 980 | 1049.7 | 1916 | 1907.3 |
| D-3-3 | 1020 | | 1993 | |
| D-7-1 | 1483 | | 2611 | |
| D-7-2 | 1495 | 1484.3 | 2600 | 2440.0 |
| D-7-3 | 1475 | | 2109 | |
| D-14-1 | 1345 | | 2421 | |
| D-14-2 | 1329 | 1184.3 | 2431 | 2456.0 |
| D-14-3 | 879 | | 2516 | |
| D-28-1 | 1482 | | 2723 | |
| D-28-2 | 1401 | 1356.7 | 2903 | 2907.3 |
| D-28-3 | 1187 | | 3096 | |

Note: A-B-C, N in A represents NSPE−ECC, D in A represents DSPE−ECC, B and C represent age and specimen number, respectively.

### 3.3. Stress−Strain Curve of Different−Aged Specimens

Figure 6 shows the stress−strain curves of NSPE−ECC and DSPE−ECC specimens at different ages during uniaxial tensile tests. From Figure 6a,e, it can be seen that both NSPE−ECC and DSPE−ECC at 3 days have already shown obvious strain hardening at the early stage of cement hydration, where the ultimate tensile strain of NSPE−ECC at 3 days was 4.4~5.1%, and the maximum strain of DSPE−ECC was only 2.78~3.48%. From the comparison of Figure 6b,c,f,g, it can be seen that the ultimate tensile strain of NSPE−ECC and DSPE−ECC at 7 d was basically unchanged as the maintenance age increased to 14 days. From Figure 6d,h, it can be seen that the ultimate tensile strains of NSPE−ECC and DSPE−ECC at 28 days had increased to 6.12~7.49% and 4.4~5.2%.

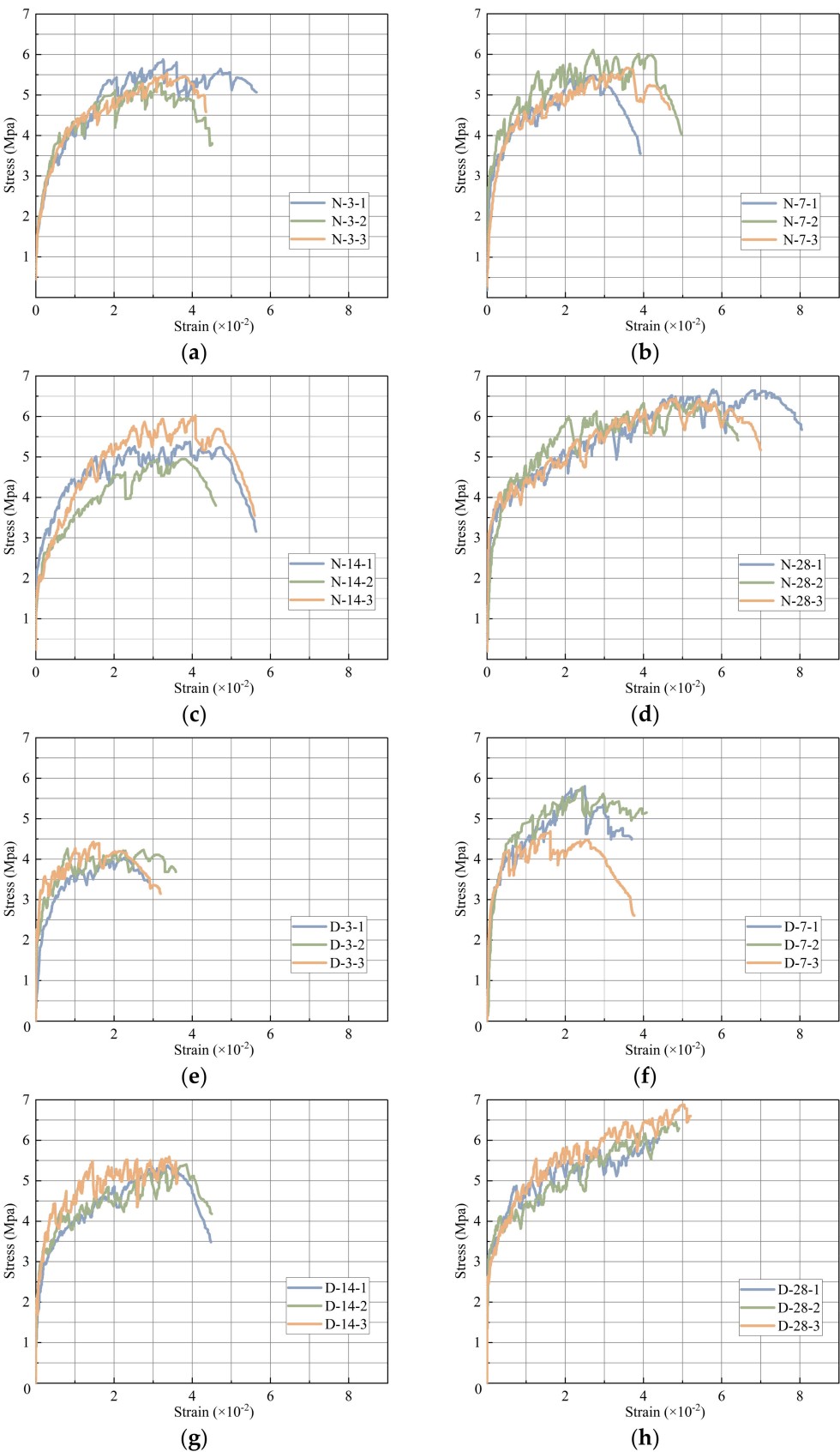

**Figure 6.** Stress−Strain curve: (**a**) 3 d, (**b**) 7 d, (**c**) 14 d, (**d**) 28 d of NSPE−ECC, (**e**) 3 d, (**f**) 7 d, (**g**) 14 d, (**h**) 28 d of DSPE−ECC.

### 3.4. Analysis of Bonding Properties

The results of the evaluation of the bonding properties of the test specimens are presented in Table 7 in accordance with the theoretical calculation formula of the bonding properties evaluation index proposed in Section 2.3 and the results of the PE−ECC uniaxial tensile test.

**Table 7.** PE−ECC bonding properties calculation results.

| Specimen Number | Bond Strength /MPa | | Average Number of Cracks /N | |
|---|---|---|---|---|
| ECC | NSPE−ECC | DSPE−ECC | NSPE−ECC | DSPE−ECC |
| 3d-1 | 2.42 | 1.78 | 8 | 5 |
| 3d-2 | 2.24 | 1.90 | 9 | 8 |
| 3d-3 | 2.29 | 1.14 | 5 | 5 |
| 7d-1 | 2.30 | 2.34 | 11 | 8 |
| 7d-2 | 2.61 | 2.51 | 12 | 11 |
| 7d-3 | 2.36 | 2.10 | 6 | 22 |
| 14d-1 | 2.34 | 2.23 | 5 | 4 |
| 14d-2 | 2.00 | 2.23 | 32 | 14 |
| 14d-3 | 2.40 | 2.38 | 5 | 20 |
| 28d-1 | 2.76 | 2.56 | 75 | 13 |
| 28d-2 | 2.72 | 2.57 | 22 | 30 |
| 28d-3 | 2.68 | 2.78 | 27 | 5 |

Note: A-B, A and B represent age and specimen number, respectively.

Table 7 shows that the bonding strength of NSPE−ECC was better than that of DSPE−ECC at 3 and 28 days, and this difference was most apparent at 3 days. At this time, the bond strength of NSPE−ECC was much higher than that of DSPE−ECC, and the data on DSPE−ECC's bond strength showed that the adhesive substances made by DSPE−ECC were not very stable. Moreover, the bond strength of DSPE−ECC was 36.11% lower than that of NSPE−ECC at 3 days. However, the bond strength of DSPE−ECC at 28 days increased by 56.51% compared with that at 3 days, which was slightly less than that of NSPE−ECC. The bond strengths of NSPE−ECC and DSPE−ECC were the same at 7 and 14 days. This finding suggests that the degree of cement hydration of NSPE−ECC and DSPE−ECC is similar at these times. In addition, the rate of bond strength growth slowed down after 7 days, which indicates that desert sand has a large impact on how cement starts to harden.

The average number of cracks in NSPE−ECC at 3 and 28 days was higher than that in DSPE−ECC at the same age, and the number of cracks in NSPE−ECC and DSPE−ECC was similar at 7 and 14 days, which suggests the two PE−ECCs did not differ appreciably at this time. However, the data was discrete, which meant that the cement matrix hydration products were unstable. Moreover, the fiber/matrix bonding properties at this time did not show many differences. A positive correlation existed between the average number of cracks and PE−ECC displacement. The calculated results in Table 7 match the displacement of NSPE−ECC in the uniaxial tensile test results, which was greater than that of DSPE−ECC. The number of cracks in DSPE−ECC at 28 days was much higher than at 3 days. This result shows that as the curing ages increase, the number of cracks in DSPE−ECC also increases. The 28th day bonding properties of DSPE−ECC were only slightly lower than those of NSPE−ECC in terms of bond strength, and the bonding properties became steadier over time.

### 3.5. Microscopic of Sands and Bonding Sections

3.5.1. Analysis Results of Desert Sand and Natural Sand

As depicted in Figures 7 and 8, SEM and XRD analyses were conducted on washed natural sand and desert sand to determine the differences in raw material properties.

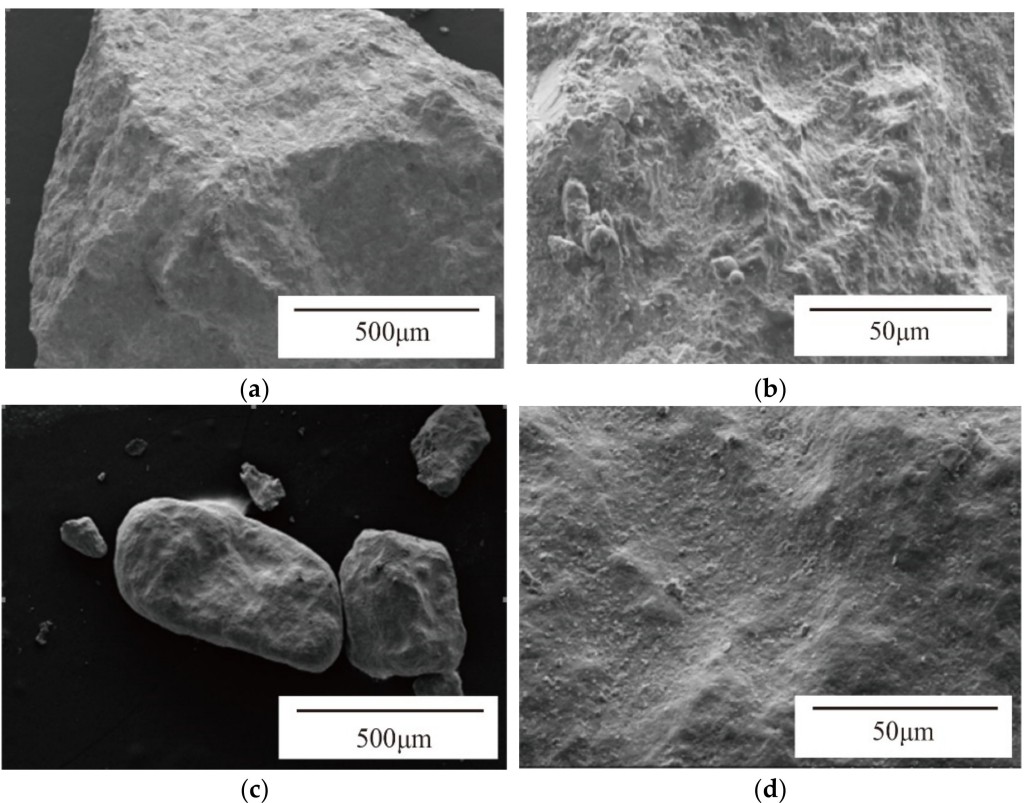

**Figure 7.** Microstructures of the specimens: (**a**) 500 μm, (**b**) 50 μm of natural Sand, (**c**) 500 μm, (**d**) 50 μm of desert sand and replacement rate.

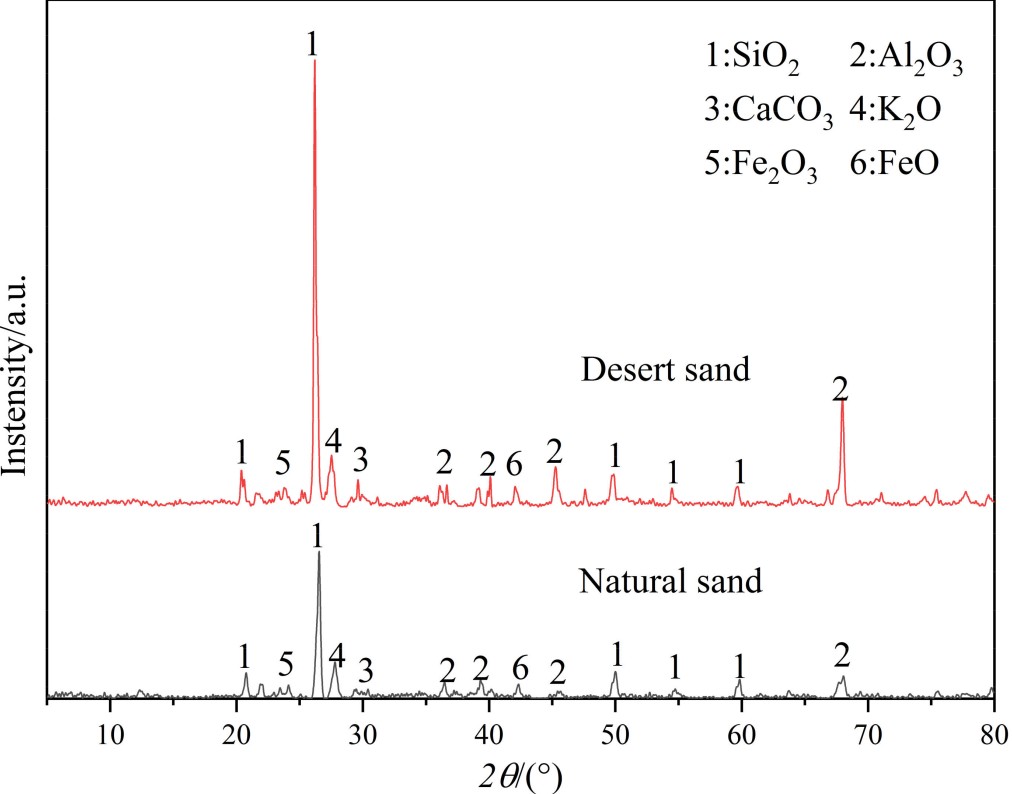

**Figure 8.** Sand diffraction diagram.

Figure 7 shows the uneven surface of natural sand, which was poured into the cement base as a fine aggregate. The rough surface would expand the reaction area when the cement hydration reaction begins to accelerate the hydration process. Therefore, the ultimate load of NSPE−ECC at 3 d was higher than that of DSPE−ECC at the same age. The uneven surface morphology provided a larger bonding surface for cement and increased the bonding strength of the cement base. Desert sand has a smaller particle size, a smoother surface, and higher water absorption compared to natural sand, reducing the cement matrix's bonding properties and the degree of cement hydration reaction. The small grain size of desert sand also filled the micropores in the cement matrix, reducing the number of micropores and enhancing the fiber/matrix interfacial adhesion so that the crack load of DSPE−ECC was higher than that of NSPE−ECC.

As shown in Figure 8, in $2\theta = 26°$ ($\theta$ is the rotation angle of the sample, and $2\theta$ is the rotation angle of the ray collected by the probe), the desert sand has a high $SiO_2$ content, indicating excellent crystallinity and hardness. Extruding the fiber roughens the fiber surface, which increases the frictional force between the fiber and matrix and prevents cracking. However, the fiber broke more easily during subsequent stresses, causing the test piece to fail; thus, DSPE−ECC had a high crack load and weak strain capacity. Desert sand was more alkaline than natural sand because $Al_2O_3$ ($2\theta = 68°$) was more abundant in alkaline oxides in the desert sand. An alkaline environment induces an alkaline aggregate reaction in the matrix, reducing the matrix's durability and cohesion.

3.5.2. Matrix Transition Zone and Hydration Product Analysis

Samples were collected from the failure surface of the tensile specimen (with many fibers attached to the surface) for SEM and XRD analyses to analyze and verify the results of the uniaxial tensile test and adhesive force analysis. The results are depicted in Figures 9 and 10.

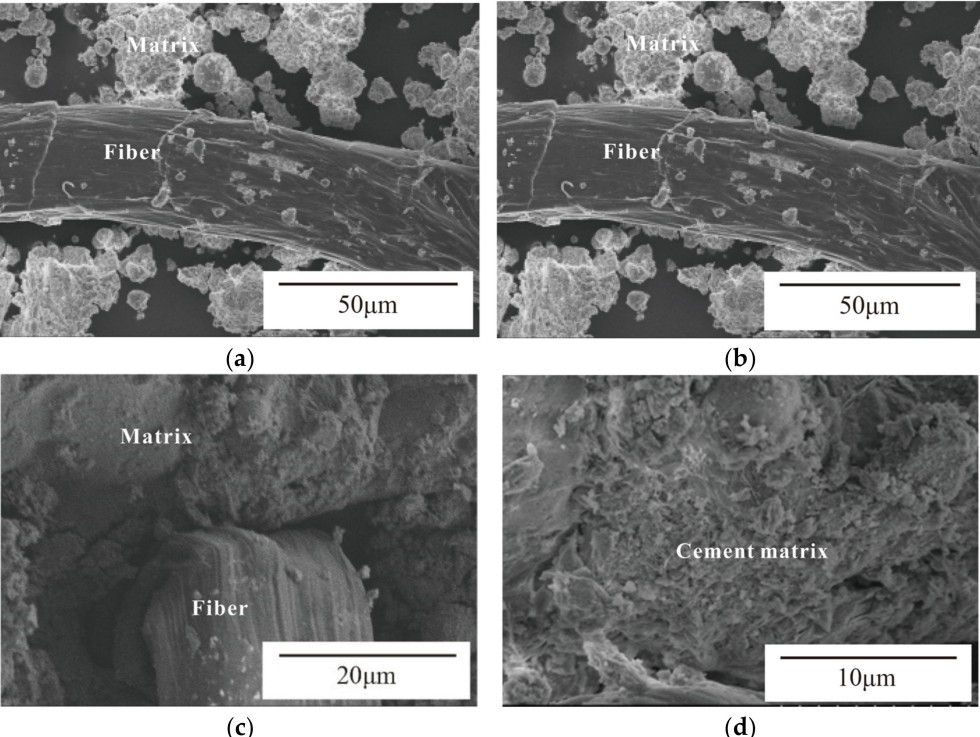

**Figure 9.** *Cont.*

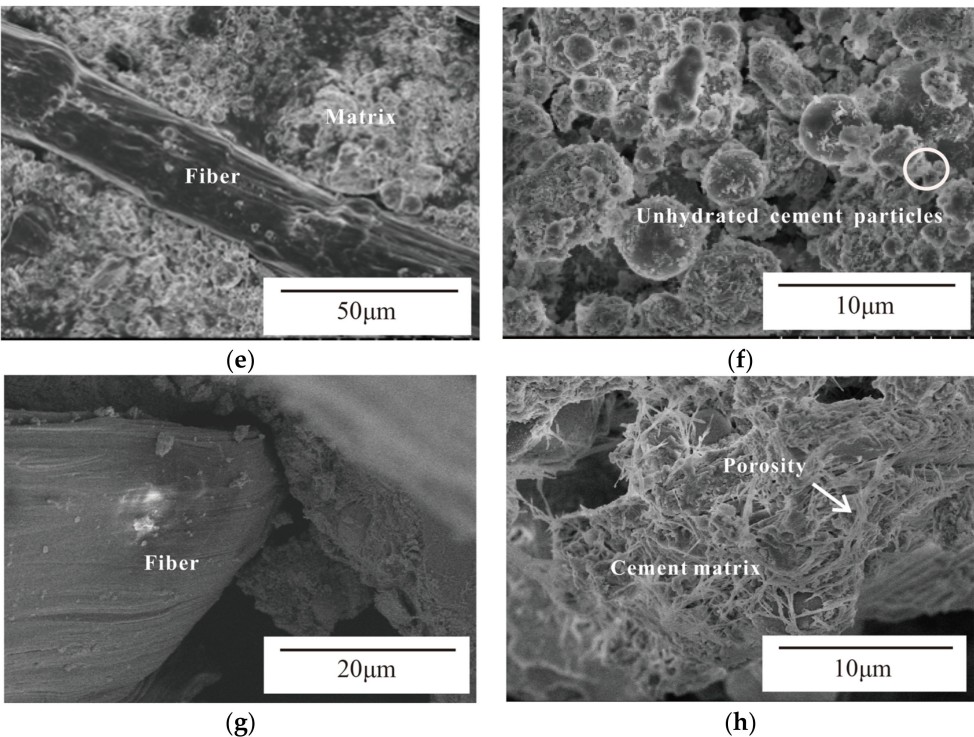

**Figure 9.** Microstructures of the specimens: (**a**) 3 d transition zone of NSPE−ECC, (**b**) 3 d matrix of NSPE−ECC, (**c**) 28 d transition zone of NSPE−ECC, (**d**) 28 d matrix of NSPE−ECC, (**e**) 3 d transition zone of DSPE−ECC, (**f**) 3 d matrix of DSPE−ECC, (**g**) 28 d transition zone of DSPE−ECC, (**h**) 28 d matrix of DSPE−ECC.

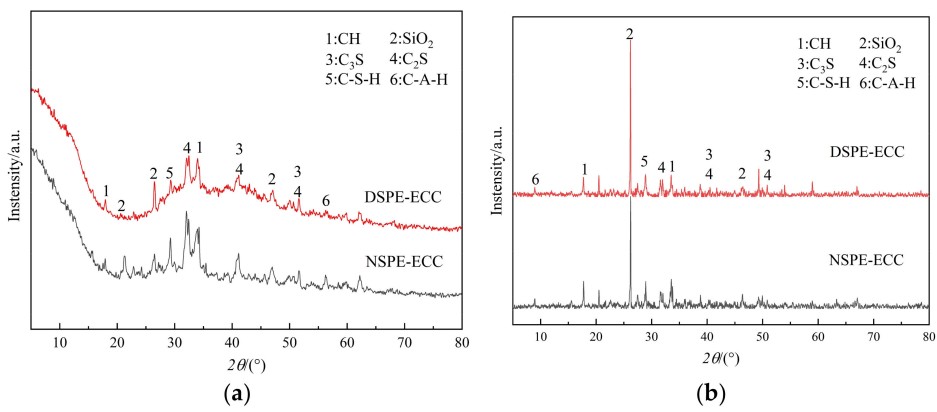

**Figure 10.** Mineral composition at different ages: (**a**) 3 d, (**b**) 28 d.

A comparison of Figure 9a,e reveals that the connection between the matrix and fiber was tighter in Figure 9e, indicating that the addition of desert sand strengthens the bond between the fiber and matrix. The DSPE−ECC matrix was dense, and the matrix strength was more remarkable, which made the DSPE−ECC crack load stronger than the NSPE−ECC crack load at the same age. However, the DSPE−ECC specimen was more susceptible to damage after the matrix was fractured owing to the smaller particle size and smooth surface of desert sand. Consequently, the ultimate load and strain of DSPE−ECC at 3 days were less than those of NSPE−ECC, as were the bonding properties. The hydration degree of NSPE−ECC and DSPE−ECC was almost the same, and a large amount of C−S−H gel was formed on the surface of the matrix and attached to the fibers, as shown in Figure 9c,g. The 28−day−old DSPE−ECC and NSPE−ECC fibers formed an excellent bridge with the matrix; therefore, the ultimate loads of DSPE−ECC and NSPE−ECC were the same.

Figure 10a showed that the proportions of C$_3$S and C$_2$S in the hydration process of the PE−ECC matrix after 3 days were greater than those of other substances. The matrix contains the C−A−H gel produced by the initial hydration of tricalcium aluminate and a small amount of CH. Figure 9b,f shows that a remarkable amount of cement raw materials need to be fully hydrated in the formed matrix, confirming that the matrix at 3 days was in the initial stage of hydration, during which NSPE−ECC was in the fiber roughness stage. The characterization effects of cement hydration, particle size, and compactness were more substantial for NSPE−ECC than for DSPE−ECC, and the bonding strength of NSPE−ECC was greater than that of DSPE−ECC.

It can be seen from Figure 9d,h that the 28−day−old NSPE−ECC hydration particles form a complete cement matrix surface, whereas the DSPE−ECC hydration particles form fibrous networks with numerous micropores. At the same time, Figure 10b shows that the diffraction results of the two PE−ECCs at 28 days were identical, indicating that their hydration and bonding properties were comparable. This suggests that the NSPE−ECC fibers are more firmly bonded to the matrix, making the fibers more challenging to extract. Consequently, DSPE−ECC's limit strain at the same age would be less than that of the NSPE−ECC.

The C−S−H gel was viscous in the solid and liquid states and could form strong bonds with fine aggregate; this property was the primary factor influencing the bonding properties between the PE−ECC fiber and matrix. The SEM test times were extended, as depicted in Figure 11, to determine the formation process of C−S−H gel in NSPE−ECC and DSPE−ECC at varying ages. The C−S−H gel of NSPE−ECC at 3 and 28 days corresponded to two forms: flocculating and block−polymerizing. The C−S−H gel of DSPE−ECC also corresponded to two conditions: flocculent and a fibrous and blocky combination. The NSPE−ECC matrix's C−S−H gel developed more quickly during the hydration process because the actual development form of C−S−H gel was from flocculent to block−like polymer. Figure 10 shows that the strengths of the DSPE−ECC's matrix C−S−H gel at 3 and 28 days were lower than those of the NSPE−ECC's matrix C−S−H gel. This finding indicates that the bonding strength of NSPE−ECC is stronger than that of DSPE−ECC, but the strength gap between the two C−S−H gel strengths is considerably reduced at 28 days. This outcome was because the desert sand's strength increased with curing ages, and the influence of water absorption saturation on cement hydration decreased, resulting in the same bonding properties for DSPE−ECC and NSPE−ECC.

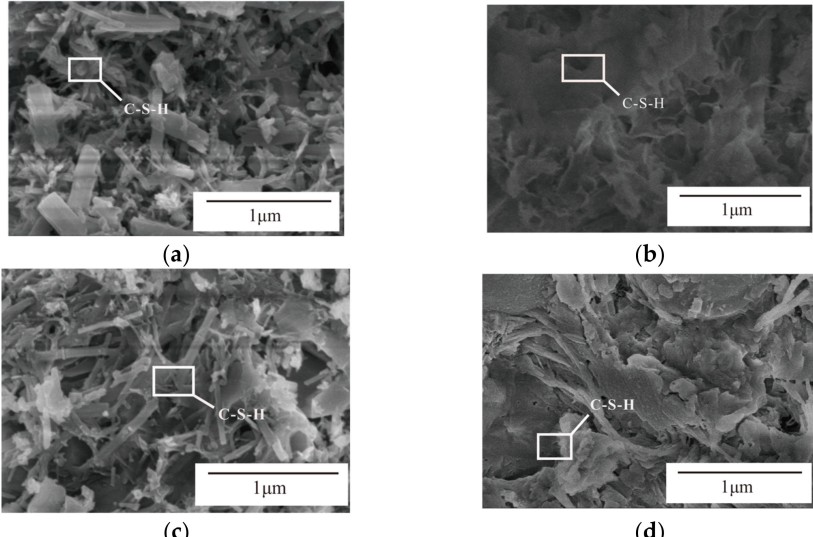

**Figure 11.** Matrix C−S−H gel history diagram: (**a**) 3 d, (**b**) 28 d of NSPE−ECC, (**c**) 3 d, (**d**) 28 d of DSPE−ECC.

Figure 12 shows that more $C_2S$ and $C_3S$ undergo hydration reactions in the absence of desert sand in the NSPE−ECC matrix to generate large and irregular flat or fiber−reticulated C−S−H gel. Moreover, the uneven appearance of the natural sand surface created a larger reaction gap for the cement, causing the generated C−S−H gel to adhere to the gap, thereby increasing the bonding area and enhancing the bonding properties of NSPE−ECC. The DSPE−ECC desert sand had a low water content and a high water absorption rate. It absorbed free water that should be involved in cement hydration, which affected the cement hydration reaction, reduced the generation of C−S−H gel in DSPE−ECC, and caused the bonding properties of DSPE−ECC to be inferior to that of NSPE−ECC.

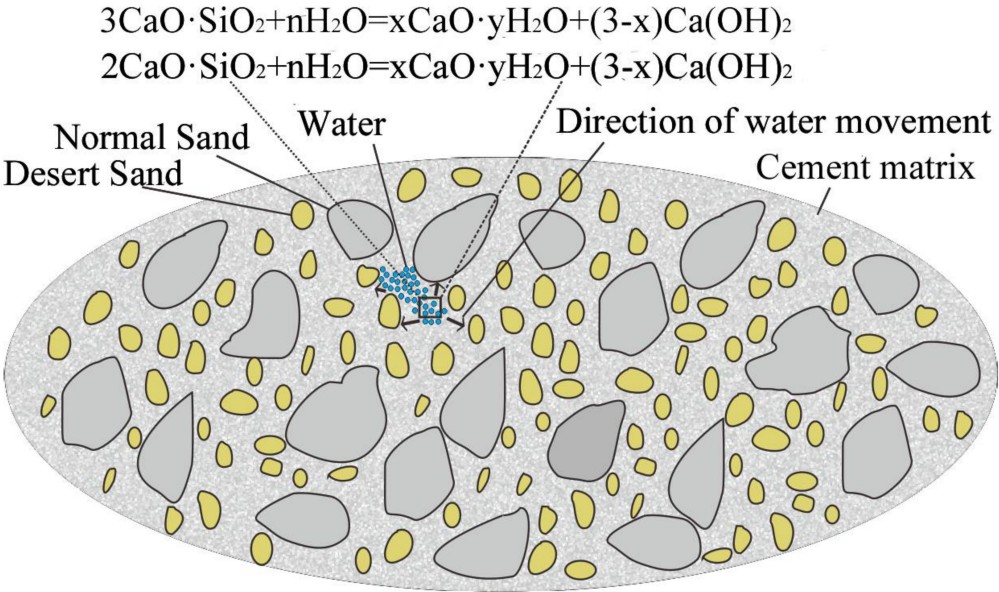

$$3CaO \cdot SiO_2 + nH_2O = xCaO \cdot yH_2O + (3-x)Ca(OH)_2$$
$$2CaO \cdot SiO_2 + nH_2O = xCaO \cdot yH_2O + (3-x)Ca(OH)_2$$

**Figure 12.** C−S−H gel generation process.

Therefore, as DSPE−ECC aged, its performance became increasingly more stable, and its ultimate tensile load was only marginally lower than that of NSPE−ECC at 28 days. The addition of desert sand also improved the fiber/matrix interface, resulting in a matrix cracking load greater than that of NSPE−ECC and the formation of ductility and multi−crack steady−state cracking. However, the high water absorption of desert sand slowed down the cement hydration process, which resulted in a distinction between the interfacial bonding properties of NSPE−ECC and DSPE−ECC at 3 and 28 days. If DSPE−ECC was cured for a longer period, its bonding properties would tend to be more stable, which has many practical applications.

## 4. Conclusions

In this paper, bonding property evaluation indexes were used to determine how the bonding properties of NSPE−ECC and DSPE−ECC changed over time by the uniaxial tensile test. The results of the calculation of the bond performance evaluation index and the results of the uniaxial tensile test were examined using microscopic testing. The main conclusions are as follows:

- The addition of desert sand enhanced the fiber/matrix's bonding properties. The cracking loads of the DSPE−ECC at 3, 7, 14, and 28 days increased by 16.72%, 28%, 23.23%, and 10.05%, respectively, compared with those of the NSPE−ECC of the same age, indicating that the addition of 30% desert sand can increase the cracking load of PE−ECC.
- It can be seen from the bond properties evaluation index that the bond strength of NSPE−ECC is better than that of DSPE−ECC at 3 and 28 days. Although the bond

strength of DSPE−ECC is 36.11% lower than that of NSPE−ECC at 3 days, at 28 days, the bond strength of DSPE−ECC increased by 56.51% compared with that at 3 days, which is only slightly less than that of NSPE−ECC. The growth rate of bond strength started to slow from 7 days. The number of cracks in DSPE−ECC at 28 days was much higher than that at 3 days, which showed that the development of multiple cracks in DSPE−ECC became better as the curing ages increased. Moreover, the bonding properties of DSPE−ECC on day 28 were slightly lower than those of NSPE−ECC in terms of bond strength, and the bonding properties became steadier over time.

- The micro−test results are consistent with the calculation results of the bonding property evaluation indexes and tensile test results at 3 and 28 days in terms of fiber roughness, matrix compactness, and hydration degree. This result shows that the calculation results of bond performance evaluation indexes can reflect the changes in the bonding properties of NSPE−ECC and DSPE−ECC.
- Desert sand slowed the rate at which the combination of $C_2S$, $C_3S$, and water molecules in the matrix combined to form C−S−H owing to its low water content and high water absorption. Therefore, the formation of C−S−H gel in the NSPE−ECC matrix was faster than that in the DSPE−ECC matrix.
- The initial tensile load of DSPE−ECC was low; however, the ultimate tensile load of DSPE−ECC increased substantially as curing ages increased, demonstrating obvious multi−crack steady−state cracking, and the bonding properties tended to stabilize, providing theoretical support for the practical engineering application of this material.

**Author Contributions:** Conceptualization, Y.N. and F.H.; methodology, Y.N.; software, Y.N. and X.Y.; validation, Y.N. and Q.L.; formal analysis, Y.N.; investigation, Y.N. and Q.L.; resources, F.H.; data curation, Y.N. and Q.L.; writing—original draft preparation, Y.N.; writing—review and editing, Y.N. and F.H.; visualization, F.H. and X.Y.; supervision, F.H.; project administration, F.H.; funding acquisition, F.H. All authors have read and agreed to the published version of the manuscript.

**Funding:** This study was financially supported by the National Natural Science Foundation, Youth Science Foundation of China (No. 51708479).

**Institutional Review Board Statement:** Not applicable.

**Informed Consent Statement:** Not applicable.

**Data Availability Statement:** All data generated and analyzed during this study are included in this article.

**Conflicts of Interest:** The authors declare no conflict of interest.

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
