# Peer review of "Effect of Desert Sand on the Section Bonding Properties of Polyethylene Fiber−Engineered Cementitious Composites"

_applsci, doi:10.3390/app13106078_

Round 1

Reviewer 1 Report

The bonding property evaluation indexes were used in this paper to determine how the bonding properties of NSPE-ECC and DSPE-ECC changed over time using the uniaxial tensile test. At the same time, the results of the bond performance evaluation index calculation and the uniaxial tensile test were examined and verified using microscopic testing. The paper sounds good and it can be publish after some minor corrections.

1. It would be better to include a table with important findings in previous published relevant work. 

2. Abstract should be concise. 

The bonding property evaluation indexes were used in this paper to determine how the bonding properties of NSPE-ECC and DSPE-ECC changed over time using the uniaxial tensile test. At the same time, the results of the bond performance evaluation index calculation and the uniaxial tensile test were examined and verified using microscopic testing. The paper sounds good and it can be publish after some minor corrections.

1. It would be better to include a table with important findings in previous published relevant work. 

2. Abstract should be concise. 

Author Response

First of all, thank you for your patient review and suggestions. Since receiving your suggestions, we have attached great importance to them and made modifications to the manuscript. We have responded to your comments in Word and thank you again for your patient review.

Reviewer 2 Report

1.      The introduction has to be strengthened, and it should address and refer to contemporary publications like:

https://doi.org/10.3390/polym14214671

https://doi.org/10.3390/app9091857

2.      Materials should be combined in precise weight percentages or atomic percentage ratios.

3.      The entire document needs to be proofread.

4.      Several typing errors need to be corrected, like the omission of spaces between the numerical value and unit, the superposition or subscription of letters, etc.

5.      Each equation that is utilized should be integrated with its source.

6.      For simple comparisons, scale bars for each image should be included, as in Figure 4.

7.      If at all feasible, the XRD analysis for all samples should be supplied since it is insufficient. For instance, the XRD charts should be used to assess the discovered phases, average crystallite size, etc.

Author Response

First of all, thank you for your patient review and suggestions. Since receiving your suggestions, we have attached great importance to them and made modifications to the manuscript. Our response to you are all in Word for your convenience in viewing.

Reviewer 3 Report

The article is ready for publication in its present form.

Author Response

First of all, thank you for your patient review, we feel honored to receive your affirmation and support, and thank you again for your patience and support.

Reviewer 4 Report

REVIEW

to the article by

Yanfeng Niu, Fengxia Han, Qing Liu and Xu Yang

“Effect of Desert Sand on the Section Bonding Properties of Polyethylene Fiber Engineered Cementitious Composites”

submitted to “Applied Sciences”

The article is made at a high scientific and experimental level.

It is of great practical importance.

There are no comments for the article.

The presented article is well illustrated and thought out.

In further studies, attention should be paid to the granulometric and mineralogical composition of aeolian sands, as was done in the article: Akulov, N.I., Rubtsova, M.N. Aeolian deposits of rift zones // Quaternary International 234 (2011) 190-201.

The article can be published in the journal "Applied Sciences".

Author Response

First of all, thank you for your patient review, we feel honored to receive your affirmation and support, we attach importance to your comments, and thank you again for your patient review and precious opinions.